# Psychosocial Factors and Sociodemographic Characteristics Associated with Suicidality Risk in Chilean Adolescents

**DOI:** 10.3390/children9081185

**Published:** 2022-08-08

**Authors:** Christianne Milena Zulic-Agramunt, Iris Paola Guzmán-Guzmán, Pedro Delgado-Floody, Monserrat Belén Cerda Saavedra, Patricio Gutierrez De La Fuente, Mario Meza Solano, Claudia Sagredo Berrios, Carles Pérez Testor

**Affiliations:** 1Departament of Mental Health and Psychiatry, Universidad de La Frontera, Temuco 4811230, Chile; 2FPCEE Blanquerna, Universitat Ramon Llull, 08022 Barcelona, Spain; 3Faculty of Chemical-Biological Sciences, Universidad Autónoma de Guerrero, Guerrero 39087, Mexico; 4Department of Physical Education, Sport and Recreation, Universidad de La Frontera, Temuco 4811230, Chile; 5Department Physical Education and Sports, Faculty of Sport Sciences, University of Granada, 18011 Granada, Spain; 6Faculty of Chemical and Pharmaceutical Sciences, Universidad de Chile, Santiago 8380000, Chile; 7Department of Psychology, Universidad de Chile, Santiago 7800284, Chile; 8Department of Pediatrics, Universidad de Chile, Santiago 9480000, Chile

**Keywords:** suicide, teenagers, self-esteem, mental health

## Abstract

Background: Suicidality in adolescents is a growing concern and is currently a public health issue in Chile and the world. Objective: To determine the association between the risk of suicidality with self-harm, sociodemographic parameters (that is, gender and type of school), psychosocial variables, and social and family support in Chilean adolescents. Methods: In a cross-sectional study, 829 (377, 45.5% girls) children/adolescents between 10 and 19 years of age participated. Suicidality, self-esteem, health-related quality of life (HRQoL), and social support perception were evaluated by standard, validated questionnaires. The presence of self-harm, dating violence, and family dysfunction was also evaluated through a self-report survey. Results: Suicidality risk was related to low self-esteem (OR = 9.73; 95%; CI = 6.62–14.28; *p* < 0.001), low HRQoL (OR = 5.0; 95%; CI = 3.51–7.13; *p* < 0.001), low social support (OR; 3.38, 95%; CI; 2.48–4.6; *p* < 0.001), and self-harm (OR = 8.03; 95%; CI = 5.69–11.33; *p* < 0.001). In family terms, suicidality risk was associated with exposure to physical (OR = 2.47, 95%CI; 1.69–3.6; *p* < 0.001) and psychological (OR = 1.78, 95; 1.33–2.39; *p* < 0.001) aggression between parents, and with considering their family dysfunctional (OR = 2.41 95%; CI = 1.69–3.41; *p* < 0.001). Finally, suicidality was associated with feeling mistreated by a boyfriend/girlfriend (OR = 2.18; 95% CI = 1.19–3.98; *p* = 0.011). Conclusion: Suicidality was associated with self-harm, low social, psychological and family well-being, and/or feeling mistreated by a boyfriend/girlfriend.

## 1. Introduction

Suicidality in adolescents is a growing concern and is currently a public health issue in Chile and the world. Suicidality captures a range of factors, from mere suicidal ideation to deliberate self-harm to the act of killing oneself, and at present, thousands of teenagers die every year by suicide [1], highlighting suicide as a significant and concerning cause of death among adolescents [2]. The literature shows that suicide in adolescents is tragic and must be actively prevented [3]. In Chile, the prevalence of attempted suicide in adolescents within the central zone (Metropolitan area) [4] and southern region (Concepción) [5] exceed international prevalence.

In addition, suicidality is a public health problem that requires attention. Suicidality risk is defined as the risk of suicide, usually indicated by suicidal ideation or intent, especially as evident in the presence of a well-elaborated suicidal plan [6]. Prediction of attempted suicide is difficult and complex, as it is based on multiple, correlated, and sometimes interacting risk factors [7], such as psychiatric, somatic, psychological, and social variables, which are considered central elements in suicide prevention strategies [8].

Self-harm also stands out as a related public health issue. In 2011, the CASE study revealed that, of 30,477 adolescents (between 14 and 17 years) in six European countries, 2.6% reported a single episode of self-harm in the last year, and 3.2% reported multiple episodes during the same period [9]. Approximately 80% of the respondents said they had not thought of self-harm in the last year. Some researchers divide self-harm into suicidal and nonsuicidal, even considering the latter a protective factor against suicidal behavior [10,11]. On the other hand, other authors suggest self-harm is a risk factor for suicidal ideation or behavior, noting that they both share relationships with similar psychiatric comorbidities and environmental risk factors (for example, history of sexual abuse, physical violence, and family dysfunction) [12,13,14]. In Chile, there are few studies on self-harm; one of these studies found that self-injury was related to suicide ideation and planning, with girls being at higher risk [4]. Nevertheless, deliberate self-harm’s relation to suicide attempts remains unclear.

In terms of the contextual predictors of suicidality, defined as a spectrum of behaviors that includes deliberately harming oneself with the intention of dying and therefore the suicidality risk [15], stressful events that involve interpersonal relations, especially family (e.g., family history with suicidal acts, familial conflict, and dysfunctional childrearing) stand out, as do relationship difficulties with peers, such as being the victim of dating or other violence, all of which emphasize negative social experiences as a predictor [16]. Another factor to consider is self-esteem, which has been linked to better adjustment and psychological functioning [17], where studies suggest that low self-esteem is a predictor of suicidal ideation as well as completed suicide [18].

Finally, few studies consider concepts from positive psychology, such as health-related quality of life (HRQoL) as predictors of suicide; one exception is a study conducted on adolescents that considered HRQoL the main mediator between psychopathology and suicidality [19]. There are more studies that focus on mental disorders than on health parameters such as well-being, a component that the HRQoL includes and that has been suggested as a protective factor against suicidality. Taken together, the objective was to determine the association between the risk of suicidality with self-harm, sociodemographic parameters (that is, gender and type of school), psychosocial variables, social and family support in Chilean adolescents.

## 2. Materials and Methods

### 2.1. Participants

A descriptive, cross-sectional study was performed on a total of 829 children-adolescents: 452 (54.5%) boys and 377 (45.5%) girls between 10 and 19 years according to the current World Health Organization (WHO) definition of adolescence [20]. The study was conducted at eight schools: five public, two subsidized, and one private. The instruments were given to all adolescents who agreed to participate (prior informed consent and parents’ consent), from courses between fifth grade elementary and the final year of high school.

The inclusion criteria were (i) belonging to an educational centre; and (ii) being aged 10 and 19 years. The exclusion criteria were (i) having a musculoskeletal disorder and (ii) any other known medical condition that might alter the participant’s response to questionnaires. Moreover, schoolchildren with physical, sensory or intellectual disabilities were excluded from this study

The study was approved by the Ethics Committee of the Metropolitan South Health Service (SSMS) and the Universidad Alberto Hurtado (UAH).

### 2.2. Instruments

To clarify the factors studied, the following questionnaires were used. The first six have been validated in Chilean samples and used in similar, previous studies [21,22,23]. These consisted of self-administered surveys. At the time of application, the adolescents were accompanied by two monitors who, although they ensured their privacy, had their support to be able to resolve any conceptual doubts that might arise. Sociodemographic parameters such as sex and type of school were also collected.

### 2.3. Suicidality and Self-Harm

The Suicidality Scale was created by Okasha et al. [15]. It is a self-report scale and is made up of four items, where the first three explore suicidal ideation and the fourth question inquiries about suicide attempts and has four possible answers: a. No attempts; b. An Attempt; c. Two attempts; d. Three or more.

The four items are Likert-type, each with four categories. The total score ranges from 0 to 12 points. Scores > 5 points define suicide risk. Cronbach’s alpha coefficient was 0.87 in a study from Chile where it was validated by Salvo [23].

In addition to the standard items, we included an additional item to investigate the presence of self-harm: “Have you ever self-harmed?”, with a dichotomous response of “Yes” or “No”. We also added the question do you consider self-harm as a problem, with a yes or no answer.

### 2.4. Psychosocial Variables

The Rosenberg Self-Esteem Scale was developed by Rosenberg to assess self-esteem in adolescents [17]. It is a self-report, Likert type, with 10 items, whose content focuses on feelings of respect and acceptance of oneself. Each item has five possible answers. The total score ranges between 10 and 50 points. The results are categorized as follows: High self-esteem from 30 to 40 points; Normal self-esteem from 26 to 29 points: Low self-esteem < 25 points. Cronbach’s alpha coefficient was 0.82 in a study from Chile in a similar population, and its items had adequate discriminative capacity [23].

The KIDSCREEN-27 is an abbreviated version of the KIDSCREEN-52 for assessing health-related quality of life (HRQoL). It has a total of 27 Likert-type items, each on a five-point scale from ‘not at all’ to ‘extremely’ or from ‘never’ to ‘always’ [22]. The scale captures five domains: physical well-being (five items) explores the level of physical activity, energy, and fitness of the child/adolescent; psychological well-being (seven items) examines the psychological well-being of the child/adolescent, including positive emotions and satisfaction with life, as well as the absence of feelings of loneliness and sadness; autonomy and parents (seven items) explores the quality of the interaction between the child/adolescent and their parents or caregivers, the family environment, family support, the perceived level of autonomy and satisfaction with economic resources; social support and peers (four items) explores the quality of the interaction between the girl/adolescent and her peers; school environment (four items) explores the child/adolescent’s perception of their own cognitive ability, learning, concentration, and their feelings towards school. Score < 4 is considered low HRQoL (moderate/low and low). Cronbach’s alpha in a study from Chile in a similar population was 0.89. [22].

### 2.5. Social and Family Support

The Perceived Social Support Scale was created by Zimet [24]. It is made up of 12 items, capturing perceived social support in three areas: family, friends, and significant others. Scores < 3 points are considered low social support. Responses are on a scale from strongly disagree to strongly agree. Cronbach’s alpha coefficient was 0.89 in study from Chile in a similar population, and its items had adequate discriminative capacity [23].

To study domestic violence, two questions were asked: (1) Have there been times when your parents have fought until they hit each other? (2) Have there been times when your parents have been offended? Each question offered an answer, yes or no. In addition, we ask this: Do you consider your family to be dysfunctional? Another question was, How many people live in your household? This was asked to find out if the family had overcrowding.

To investigate dating violence, this was asked: Have you felt mistreated by your boyfriend/girlfriend?

### 2.6. Procedure

Participants’ schools were contacted by the investigator through the network registry of the communities. For those schools that agreed to participate, a letter was sent explaining the research project, which was signed by the principal in cases where the principal gave their authorization. Enclosed with the letter was informed consent forms for participants and their parents, where it was stressed that only the students who had previously submitted both documents could take part. A presentation was given to the principal and school guidance counselor/psychologist, where the project was explained, and we planned the time and place that the students would participate. The questionnaires were completed via Survey Monkey on school computers, using unique links created for each student. Although completed on the computer, the study took place in-person and was supervised by a pediatric and adolescent psychiatrist, a psychologist, and trained interns from health programs. Every effort was made to ensure confidentiality.

### 2.7. Data Analysis

Data analysis was done in STATA v.15.0. (StataCorp, College Station, TX, USA). Normal distribution was tested using the Shapiro-Wilk test. The dichotomous variables are reported as number (n) and proportions (%), and were compared using the Chi-squared test. The Likert-based variables are reported as means and standard deviation, and compared between groups with Student’s t-test. The linear relation between the suicidal ideation with the self-esteem, HRQoL dimensions and social support was evaluated and described with Pearson’s correlation coefficient. The association between suicidal risk (i.e., suicidal ideation and suicide attempts) with sociodemographic self-harm, psychosocial variables and social–family support including dating violence was evaluated by simple logistic regression, and described with odds ratio (OR) and respective 95% confidence intervals (95% CI). For all the analyses, *p* < 0.05 was considered statistically significant.

## 3. Results

Table 1 compares the study’s variables by sex. Boys showed significantly higher self-esteem than girls (boys: 29.54 ± 5.68 score vs. girls: 28.32 ± 6.42 score; *p* 0.004). Similarly, girls had higher proportional representation in the category of low self-esteem (37.33%; *p* < 0.001). The boys also showed higher HRQoL (boys; 102.76 ± 15.91 vs. girls; 96.13 ± 17.16; *p* < 0.001). In relation to suicidal ideation, the girls scored higher than the boys (boys scoring 2.06 ± 2.73 vs. girls scoring 3.55 ± 3.51; *p* < 0.001).

Table 2 shows the frequency of students’ reporting self-harm and exposure to various problematic relationships. A higher percentage of girls reported self-harm than did boys (53.54%; *p* < 0.001).

Table 3 shows the correlation between suicidal ideation scale score and various measures of individual and social differences. Social support (*r* = −0.35, *p* < 0.005), self-esteem (*r* = −0.51, *p* < 0.001), and quality of life (*r* = −0.55, *p* < 0.001) all showed a negative correlation with suicidality. Psychological well-being showed an especially strong negative correlation with suicidality (*r* = −0.61, *p* < 0.001). These correlations maintained significance within both girls’ and boys’ groups.

Table 4 shows the association between suicidality risk category with the different parameters. Students with scores greater than five points were categorized as suicidality risk, and students with ≤5 points represent the reference category. In relation to the sociodemographic variables, suicidality risk was positively related to being over 14 years of age (OR = 1.42; 95% CI = 1.06–1.89; *p* = 0.019), being a girl (OR = 2.22, 95% CI = 1.66–2.96, *p* < 0.001), and being in a public school compared to a private school (OR = 2.0; 95% CI = 1.32–3.03; *p* = 0.001). In relation to the psychological and social variables, suicidality risk was related to low self-esteem (OR = 9.73; 95% CI = 6.62–14.28; *p* < 0.001), low HRQoL (OR = 5.0; 95% CI = 3.51–7.13; *p* < 0.001), low social support (OR; 3.38, 95% CI; 2.48–4.6, *p* < 0.001), and greater self-harm (OR = 8.03; 95% CI = 5.69–11.33; *p* < 0.001). In the family dimension, suicidality risk was associated with exposure to physical (OR = 2.47, 95%CI; 1.69–3.6, *p* < 0.001) and psychological (OR = 1.78, 95; 1.33–2.39, *p* < 0.001) aggression between parents, and with considering their family dysfunctional (OR = 2.41 95%CI; 1.69–3.41, *p* < 0.001). Finally, suicidality was associated with feeling mistreated by a boyfriend/girlfriend (OR = 2.18; 95% CI = 1.19–3.98; *p* = 0.011).

Table 5 shows variables’ association with suicide attempts, captured by item #4 on the suicidality scale (OKASHA). Self-harming, being over 14 years old, being in a public school, being a girl, having low self-esteem, low HRQoL (quality of life), low social support, exposure to physical and psychological aggression between parents, having a dysfunctional family, and mistreatment by a boyfriend/girlfriend all presented significant associations with suicide attempts. In addition, suicidality reported association with self-harm (OR; 8.03, 95% CI; 5.69–11.33, <0.001) and with self-harming ≥ 2 times (OR; 32.5 (15.0–70.6, <0.001).

## 4. Discussion

The objective of the present study was to determine the association between the risk of suicidality with self-harm, sociodemographic parameters (that is, gender and type of school), psychosocial variables, and socio-family support in Chilean adolescents.

In the present study, suicidality risk was associated with self-harm. Suicidality showed higher prevalence in girls, even higher than found in other studies, including studies conducted in Chile [4]. Although only a few studies have examined adolescent suicidality’s association with various suicide variables, including ideation, planning, and attempts, the results have varied across studies. To date, the research has demonstrated a high rate of co-occurrence between nonsuicidal self-harm and suicidal behavior in adolescents and young adults, especially girls. Indeed, studies of adolescent suicidality’s relationship to sex, age, and education level, as well as with use of mental health services, remains scarce [13]. Consequently, it is possible that the risk of suicidality in adolescents who self-harm is underestimated in some studies, implying that researchers and practitioners should be cautious about the definition of nonsuicidal self-harm. The empirical evidence shows that deliberate self-harm and suicidal ideation frequently co-occur in individuals, and share similar psychiatric comorbidities (with, for example, depression, borderline personality disorder, substance abuse, post-traumatic stress disorder, impulsivity, attention deficit, behavioral disorder) and environmental factors (for example, history of sexual abuse and physical violence, and family dysfunction) [12,13,25,26,27,28].

The second critical finding of our study was that self-harm was associated with having two or more suicide attempts, similar to what is indicated by Barreto Carvalho et al. [29] and Guerreiro et al. [30]. The former author mentions that, following suicidal ideation, the frequency of deliberate self-harm is the strongest predictor of suicide. Meanwhile, the latter author supports this finding by showing that a subgroup of adolescents that repeatedly self-harmed presented higher rates of anxiety and depression symptoms, which are strongly related to suicidality.

These results also fit with The Gateway Theory [10,31], which proposes the concept of a continuum or spectrum, considering deliberate self-harm at one end and suicide at the other, with self-harm acting as a risk factor that represents the ‘gateway’ to suicide. The evidence supports this hypothesis [32,33], emphasizing the strong co-occurrence between self-harm and suicidal behavior [34]. This invites us to consider self-harm as a critical risk factor in suicide prevention, in contrast to considering self-harm as something relatively benign, or only as a call for attention [32]. In addition, most previous studies have used samples from clinics and hospitals and not the general population, which limits their extrapolation. McManus et al. [35], for example, found that self-reported lifetime nonsuicidal self-harming increased from 2.4% in 2000 to 6.4% in 2014 in England, but most did not make use of medical or psychological services.

In this study, low self-esteem and a low quality of life were related to suicidality risk. In this vein, self-esteem is usually defined as the assessment an individual makes about themselves, which is partly determined by how the individuals believe they are perceived or evaluated by other significant people. Consequently, perceived ‘negative’ feedback from others may be detrimental to self-esteem. The nature of the relationship between self-esteem and depression, however, remains unclear [36]. More recent studies have shown a relationship between self-esteem and suicidal ideation, independent of depression and hopelessness [37,38]. Our findings support the idea of continuing to consider self-esteem to be related to suicidality, particularly but not only in girls, which is consistent with other literature for this age group [23,37]. With respect to quality of life, defined as ‘perception of the individual of their position in life in the context of the culture and values system in which they live and in relation to their objectives, expectations, standards and concerns’, girls presented the lowest levels, a result that is consistent with the literature [39]. Low quality of life has also appeared, construed as psychological well-being, as a relevant mediator between psychopathology and suicidality [18]. Although our findings are consistent with these findings in the literature [39], there remains a paucity of studies that address how positive psychological variables such as quality of life and well-being are associated with suicidality.

The results revealed an association between suicidality and low perceived social support as well as exposure to familial conflict and dysfunction, also consistent with the literature [21,23], where social support has been described as a predictor of suicidality. Perceived social support is closely linked to interpersonal relations, with both family and peers. In this respect, it is known that bullying is associated with suicidal ideation and behavior [40]. On the other hand, studies show family dysfunction is related to adolescents’ greater behavioral problems/defiance and substance use, which may in turn relate to symptoms of depression and suicidal ideation, noting that the direct relation between family dysfunction and suicidality is less clear [41]. To this end, our results suggest that exposure to familial physical and psychological aggression and perceiving one’s family as dysfunctional were associated with greater suicidality. This suggests it may be worthwhile to consider the family as a center of intervention for suicide prevention in this age group.

The last finding worthy of note was the association with feeling mistreated by a boyfriend/girlfriend and suicidality, although other research on this is sparse. Nevertheless, this idea is indirectly supported by research that self-harm, victimization by classmates, difficulty in regulating emotions, and poor communication skills are related to an increased likelihood of suffering interpersonal and partner violence [42]. If we intertwine the previously mentioned factors, our results agree with the literature indicating that adolescents who self-harm often lack strategies to cope with difficulties and tend to express disinterest and avoidance behaviors. All these factors, when taken altogether, predispose a person to a greater risk of having defective interpersonal relations [43].

### Limitations

As a cross-sectional and prospective study, it is difficult to determine causality between the factors studied and suicidality; however, the associations we found are consistent with the literature and with different models related to suicidality. Unfortunately, self-harm was only measured with a question that does not measure temporality, and it was also not a validated questionnaire, leaving it subject to the adolescent’s interpretation of what they understand self-harm to be. In this question, a history of self-harm was not considered and could be possible overlap between self-harm and risk of suicide. It would also have been advisable to include the question, “Have you ever self-harmed, with no intention of dying?”, to make it possible to delve into the phenomenon of nonsuicidal self-harm.

On the other hand, regarding dating violence, the perception of feeling mistreated by a boyfriend/girlfriend could only be measured with one question, and not with a validated test that measures dating violence, because it was not possible to find a validated questionnaire to study adolescent dating violence in Chile.

## 5. Conclusions

Suicidality is associated with self-harm, low self-esteem and quality of life, with low perceived social support, feeling mistreated by a boyfriend/girlfriend and with family dysfunction. It is recommended to be sensitive to the presence of self-harm given its strong association with suicidality, and to increase the number of prospective studies that include contextual factors, including interpersonal ones, as these may be important risk factors to consider in public policies aimed at promoting and preventing suicidality, and help to avoid reaching a point where psychopathology is already installed.

## Figures and Tables

**Table 1 children-09-01185-t001:** Parameters evaluated in the study population, by sex.

Variables	Total N = 829	Men N = 452 (54.5%)	Women N = 377 (45.5%)	*p* Value
Age (years) ^a^	13.9 ± 2.4	13.9 ± 2.41	14.0 ± 2.45	0.43
Type of school ^b^, n (%)				0.62
Private	164 (19.78)	88 (19.47)	76 (20.16)	
Subsidized	296 (35.71)	168 (37.17)	128 (33.95)	
Public	369 (44.51)	196 (43.36)	173 (45.89)	
Self-esteem ^a^	28.99 ± 6.05	29.54 ± 5.68	28.32 ± 6.42	0.004
Self-esteem ^b^, n (%)				<0.001
High	379 (47.73)	226 (51.95)	153 (42.62)	
Moderate	175 (22.04)	103 (23.68)	72 (20.06)	
Low	379 (30.23)	106 (24.37)	153 (37.33)	
Health-related quality of life ^a^	99.7 ± 16.81	102.76 ± 15.91	96.13 ± 17.16	<0.001
Physical well-being ^a^	16.3 ± 3.51	17.07 ± 3.28	15.37 ± 3.57	<0.001
Psychological well-being ^a^	25.98 ± 6.13	27.1 ± 5.58	24.65 ± 6.5	<0.001
Autonomy and parents ^a^	26.03 ± 5.78	26.65 ± 5.61	25.3 ± 5.9	<0.001
Peers and social support ^a^	16.81 ± 3.36	16.89 ± 3.26	16.71 ± 3.47	0.43
School environment ^a^	14.6 ± 2.99	14.66 ± 2.98	14.54 ± 3.01	0.56
Suicidality ^a^				
Suicidal ideation ^a^	2.74 ± 3.19	2.06 ± 2.73	3.55 ± 3.51	<0.001
Suicidality risk ^a^	3.05 ± 3.18	2.55 ± 3.01	3.64 ± 3.28	<0.001
Social support ^a^	37.58 ± 8.19	37.78 ± 7.95	37.33 ± 8.58	0.43
Social support ^b^, n (%)				0.60
Low social support	281 (35.44)	151 (34.63)	130 (36.41)	
High social support	512 (64.56)	285 (65.37)	227 (63.59)	

Data shown represent mean ± SD, and n (%). Note: *p* value < 0.05 is considered statistically significant; ^a^ student t-test. ^b^ = Chi ^2^ test.

**Table 2 children-09-01185-t002:** Frequency of social–family and individual patterns in the sample, by sex.

Variables	Total N = 829 (%)	MenN = 452 (%)	Women N = 377 (%)	*p* Value
Self-harm, yes n (%)	222 (26.78)	88 (19.47)	134 (35.54)	<0.001
Parental physical aggression, yes n (%)	132 (15.92)	64 (14.16)	68 (18.04)	0.12
Parental psychological aggression, yes n (%)	310 (37.39)	158 (34.96)	152 (40.32)	0.11
Relationship, yes n (%)	609 (74.72)	318 (72.11)	291 (77.81)	0.06
Aggression in dating, yes n (%)	46 (6.28)	29 (7.27)	17 (5.11)	0.23
Considers aggression a problem, yes n (%)	193 (23.28)	114 (25.22)	79 (20.95)	0.14

Data shown represent n (%). Note: *p* value < 0.05 is considered statistically significant; Chi ^2^ test.

**Table 3 children-09-01185-t003:** Correlation between suicide risk and other factors, also within sex.

	Total	Men N = 452	Women N = 377
Age (years)	0.14 (<0.001)	0.15 (0.001)	0.13 (0.008)
Self-esteem	−0.51 (<0.001)	−0.47 (<0.001)	−0.53 (<0.001)
HRQoL (total)	−0.55 (<0.001)	−0.47 (<0.001)	−0.58 (<0.001)
Physical well-being	−0.38 (<0.001)	−0.37 (<0.001)	−0.33 (<0.001)
Psychological well-being	−0.61 (<0.001)	−0.53 (<0.001)	−0.64 (<0.001)
Autonomy and parents	−0.36 (<0.001)	−0.26 (<0.001)	−0.41 (<0.001)
Peers and social support	−0.23 (<0.001)	−0.24 (<0.001)	−0.23 (<0.001)
School environment	−0.40 (<0.001)	−0.25 (<0.001)	−0.40 (<0.001)
Social support	−0.35 (<0.005)	−0.27 (<0.005)	−0.44 (<0.001)

Note: *p* value < 0.05 is considered statistically significant; Pearson’s r (p). HRQoL = Health-related quality of life.

**Table 4 children-09-01185-t004:** Variables associated with suicidality risk in Chilean adolescent.

	Suicidality Risk		
Variables	No (≤5 Points) * N = 535 (64.54)	Suicidality Risk (>5 Points) N = 294 (35.46)	OR (95% CI), *p* Value
Age			
≤14 Y	340 (63.55)	162 (55.1)	Comparison
>14 Y	195 (36.45)	132 (44.9)	1.42 (1.06–1.89), 0.019
Sex			
Men	329 (61.50)	123 (41.84)	Comparison
Women	206 (38.5)	171 (58.16)	2.22 (1.66–2.96), <0.001
Type of school			
Private	125 (23.36)	39 (13.27)	Comparison
Subsidized	183 (34.21)	113 (38.44)	1.97 (1.28–3.04), 0.002
Public	227 (42.43)	142 (48.30)	2.0 (1.32−3.03), 0.001
Self-harm			
No	469 (87.66)	138 (46.94)	Comparison
Yes	66 (12.34)	156 (53.06)	8.03 (5.69–11.33), <0.001
Self-esteem High	321 (62.57)	58 (20.64)	Comparison
Moderate	105 (20.47)	70 (24.91)	3.68 (2.44–5.57), <0.001
Low	87 (16.96)	153 (54.45)	9.73 (6.62–14.28), <0.001
HRQoL			
High	261 (48.79)	47 (15.99)	Comparison
Low	274 (51.21)	247(84.01)	5.0 (3.51–7.13), <0.001
Social Support			
High	384 (74.42)	128 (46.21)	Comparison
Low	132 (25.58)	149 (53.79)	3.38 (2.48–4.6), <0.001
Physical aggression of parents			
No	474 (88.6)	223 (75.85)	Comparison
Yes	61 (11.4)	71 (24.15)	2.47 (1.69–3.6), <0.001
Psychological aggression of parents			
No	361 (67.48)	158 (53.74)	Comparison
Yes	174 (32.52)	163 (46.26)	1.78 (1.33–2.39), <0.001
Dysfunctional family			
No	418 (84.1)	191 (68.71)	Comparison
Yes	79 (15.9)	87 (31.29)	2.41 (1.69–3.41), <0.001
Aggression in dating			
No	444 (95.48)	242 (90.64)	Comparison
Yes	21 (4.52)	25 (9.36)	2.18 (1.19–3.98), 0.011

Note: Data shown represent n (%) in each category, along with the odds ratio (OR) and respective 95% confidence intervals derived from a logistic regression. * Represent reference category low suicidality risk (≤5 points). A *p* value of <0.05 is considered statistically significant. HRQoL = health related to quality of life.

**Table 5 children-09-01185-t005:** Variables associated with attempted suicide in Chilean adolescents.

	Suicide Attempt	
Variables	One Attempt n = 88 (10.6)	Two or More Attempts n = 58 (7.0)
	OR (95% CI), *p* Value	OR (95% CI), *p* Value
Age > 14 y	1.76 (1.13–2.76), 0.012	1.57 (0.92–2.69), 0.09
Girls	2.38 (1.50–3.76), <0.001	3.45 (1.92–6.19), <0.001
Public school	2.51 (1.23–5.12), 0.011	5.24 (1.61–17.0), 0.006
Self-harm	12.42 (7.52–20.5), <0.001	32.5 (15.0–70.6), <0.001
Low self-esteem	3.71 (2.19–6.28), <0.001	12.27 (4.83–31.1), <0.001
Low HRQoL	5.25 (2.74–10.07), <0.001	10.14 (3.63–28.32), <0.001
Low social support	2.74 (1.74–4.33), <0.001	4.55 (2.48–8.35), <0.001
Physical aggression of parents	4.27 (2.62–6.97), <0.001	2.48 (1.32–4.67), 0.005
Psychological aggression of parents	2.22 (1.42–3.48), <0.001	2.22 (1.30–3.82), 0.004
Dysfunctional family	2.51 (1.54–4.10), <0.001	2.34 (1.29–4.23), 0.005
Aggression in relationship	1.33 (0.54–3.29), 0.53	3.12 (1.35–7.17), 0.007

Note: Data shown represent odds ratio (OR) and respective 95% confidence intervals. *p*-value < 0.05 is considered statistically significant. Reference category without suicide attempts. HRQoL= health related to quality of life.

## Data Availability

Not applicable.

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
