# Peer review of "Psychosocial Factors and Sociodemographic Characteristics Associated with Suicidality Risk in Chilean Adolescents"

_children, 2022, doi:10.3390/children9081185_

Round 1
Reviewer 1 Report
The manuscript addresses a relevant public health issue, considers a large sample of adolescents, and explores a series of predictors related to psychosocial, demographic, and risk behaviors (i.e., self-injury) that allow explore a wide range of risk factors for suicidal behavior. However, the paper needs to be improved in formal aspects (wording, grammatical errors) and in the presentation of the results. Regarding this last point, the authors only considered univariate logistic regression analyses to explore the odd ratio of suicidal behavior. While these results are relevant and informative, it is necessary to add multivariate logistic regression analyses for further exploration onf the data. Such analyses can provide more precise information about which predictors are more robust when controlling for the other variables in the same analysis. These analyses could improve considerably this paper. Therefore, it is recommended that these improvements be made to the manuscript before publication.
Section-by-section comments are detailed in the attached file

Author Response
RESPONSE TO REVIEWER
Reviewer report (1)
- I found the paper clear and well structured. The topic is of great interest and relevance to
the field.
Response: Dear Reviewer many thanks for the opportunity of the article correction, your suggestions have improved the quality of the paper.
I suggest some aspects that can be improved.
- Introduction: if the authors differ “suicidality” from “suicide risk” (line 48), why do they talk
about “suicidality risk” when introducing the aim of the work (line 80)? It could be quite
confusing.
Response: Suicidality is defined as deliberately harming oneself with the intention of dying, including what is the risk of suicide. There would be no problem in changing in line 80, the word “suicidality risk” by “suicide risk”. We use the term suicide, since the Okasha instrument uses it, but there would be no problem changing it.
Anyway, we added in the introduction, so that the concepts are clearer:
In addition, suicidality is a public health problem that requires attention. Suicidality risk is defined as: ‘a predictive statement of the probable occurrence of a fatal suicide attempt and may be conceived in terms of a complex equation’
- I think it would be clearer if the authors would follow the same order and use the same
definitions when listing variables associated with suicidality risk: (1) at the end of the
introduction (lines 80-82); (2) when describing the instruments in the Methods section
(from line 98); (3) at the beginning of the discussion (lines 223-226)
Response: Thank you very much for the suggestion, we ordered it as you suggested.
We have ordered the objectives according to your request and according to the order of materials and methods, in the same way:
...to determine the association between suicide risk with self-harm, psychosocial variables, socio-family support and sociodemographic parameters (sex, type of school) in Chilean adolescents.
- Methods:
(1) When introducing the Instruments (lines 99-100), it should be added the mention to
the self-report survey, as well as done in the abstract (line 29)
Response: Thanks a lot for the suggestion. We have added:
These consisted of self-administered surveys. At the time of application, the adolescents were accompanied by two monitors who, although they ensured their privacy, had their support to be able to resolve any conceptual doubts that might arise. Sociodemographic parameters such as sex and type of school were also collected.
(2) The way of assessing the presence or absence of Self-harm is weak because a history of
self-harm could be considered also in the case of a single episode of self-injury. I think it
could be somehow “non-specific” and it should be specified in the limitations.
Response: Thank you very much for the suggestion. We have added in limitations section:
In this question a history of self-harm was not considered and could be possible overlap between self-harm and risk of suicide. It would also have been advisable to include the question “Have you ever self-harmed, with no intention of dying?", to make it possible to delve into the phenomenon of non-suicidal self-harm.
(3) I think the question “Do you consider your family dysfunctional?” (line 141) is quite
difficult to be fully understood by children and adolescents.
Response: We understand your observation, however we took the precaution that all adolescents had two monitors while they answered the survey to ask any questions they had while they made their answers and none indicated difficulties in this regard. We added this reflected in the methodology after your suggestion.
(4) The authors did not mention the assumptions for the applicability of the t-test
(Normality, homogeneity of variance). Did they evaluate them?
Response: thank you, we have added:
Normal distribution was tested using the Shapiro-Wilk test
- Results:
(1) Table 2: “Considers aggression a problem”: I do not find this question in the Methods
section but only “Have you felt mistreated by your boyfriend/girlfriend?”. I think that
the list of questions for the self-report survey should be completed.
Response: we committed a mistake, we have added:
To investigate dating violence we asked, ‘Have you felt mistreated by your boy-friend/girlfriend?’, Considers aggression a problem? each one with a yes or no response.
(2) Where was “impulsivity” (line 185) measured? On which scale? I do not find it
mentioned.
Response: sorry about that, we committed a mistake, in a previous peer reviewed report this variable was deleted. We have changed in text.
Thank you very much for your suggestions.
All the best

Reviewer 2 Report
well written; interesting use of family variables; paper does what it proposes.
Author Response
Thank you very much for the congratulations.
All the best

Reviewer 3 Report
- I found the paper clear and well structured. The topic is of great interest and relevance to the field.
I suggest some aspects that can be improved.
- Introduction: if the authors differ “suicidality” from “suicide risk” (line 48), why do they talk about “suicidality risk” when introducing the aim of the work (line 80)? It could be quite confusing.
- I think it would be clearer if the authors would follow the same order and use the same definitions when listing variables associated with suicidality risk: (1) at the end of the introduction (lines 80-82); (2) when describing the instruments in the Methods section (from line 98); (3) at the beginning of the discussion (lines 223-226)
- Methods:
(1) When introducing the Instruments (lines 99-100), it should be added the mention to the self-report survey, as well as done in the abstract (line 29)
(2) The way of assessing the presence or absence of Self-harm is weak because a history of self-harm could be considered also in the case of a single episode of self-injury. I think it could be somehow “non-specific” and it should be specified in the limitations.
(3) I think the question “Do you consider your family dysfunctional?” (line 141) is quite difficult to be fully understood by children and adolescents.
(4) The authors did not mention the assumptions for the applicability of the t-test (normality, homogeneity of variance). Did they evaluated them?
- Results:
(1) Table 2: “Considers aggression a problem”: I do not find this question in the Methods section but only “Have you felt mistreated by your boyfriend/girlfriend?”. I think that the list of questions for the self-report survey should be completed.
(2) Where was “impulsivity” (line 185) measured? On which scale? I do not find it mentioned.

Author Response
RESPONSE TO REVIEWER
Reviewer report (3):
The manuscript addresses a relevant public health issue, considers a large sample of adolescents, and explores a series of predictors related to psychosocial, demographic, and risk behaviors (i.e., self-injury) that allow explore a wide range of risk factors for suicidal behavior. However, the paper needs to be improved in formal aspects (wording, grammatical errors) and in the presentation of the results. Regarding this last point, the authors only considered univariate logistic regression analyses to explore the odd of suicidal behavior. While these results are relevant and informative, it is necessary to add multivariate logistic regression analyses for further exploration of the data. Such analyses can provide more precise information about which predictors are more robust when controlling for the other variables in the same analysis. These analyses could improve considerably this paper. Therefore, it is recommended that these improvements be made to the manuscript before publication.
Dear Reviewer, many thanks for the opportunity of the article correction, your suggestions have improved the quality of the paper.
Section-by-section comments are detailed below:
Abstract.
- Only report CIs of self-harm. Include also in the others predictors reported in the abstract.
Response: Thank you very much for your suggestion. We have added CI in all variables:
Suicide risk was related to low self-esteem (OR = 9.73; 95% CI = 6.62-14.28; p <0.001), low HRQoL (OR = 5.0; 95% CI = 3.51-7.13; p < 0.001), low social support (OR; 3.38, 95% CI; 2.48-4.6, p < 0.001), and self-harm (OR = 8.03; 95% CI = 5.69-11.33; p <0.001). In family terms, suicidality risk was associated with exposure to physical (OR=2.47, 95%CI; 1.69-3.6, p <0.001) and psychological (OR=1.78, 95; 1.33-2.39, p<0.001) aggression between parents, and with considering their family dysfunctional (OR=2.41 95%CI; 1.69-3.41, p <0.001).
Introduction.
- Third paragraph. “Approximately 80% of the respondents said they had not had thoughts of self-harm in the last year”.
This sentence has a grammatical error. Revise and edit in the correct way.
Response: Thank you very much for your suggestion. We corrected it.
- Third paragraph. “In Chile, there are few studies on self-harm, but one by Barroilhet et al. 2012 stands out, where they describe and self-harm as being related to suicide ideation and planning, with girls being at higher risk”.
Omit unnecessary words and avoid excessive nominalization. For example: “One of these studies found that self-injury was related to suicide ideation and planning, with girls being at higher risk (Barroilhet et al., 2012)”
Response: Thank you very much for your suggestion. We have corrected it.
Materials and Methods.
- Participants: Why did you considerer the range of adolescent between 10-19 years (usually is considered 12-18).
Response: Currently, the OMS considers adolescence to be the period between 10 and 19 years of age. In addition, clinically there are teen's problems from that age and since there is a few research in this regard, there is a large gap for the population between 10 and 14 years old, which is not good, since there are very few research works that can guide public policies or clinical decisions for them.
- Instruments: In Suicidality and self-harm Section, cites the authors of "The Suicidality Scale" as Okasha et al., however, does not add the reference of these authors.
Please, add the reference.
Response: Thank you very much for your suggestion. We added it.
- Physical and mental health Section.
The end of the paragraph of this section reads: “Score <4 era considered low HRQoL…”.
Change the word "era" to its English translation.
Response: Thank you very much for your suggestion. We corrected the translation.
- Physical and mental health Section.
This section includes two instruments. The second instrument is KIDSCREEN-27. However, at the end of the paragraph you add the abbreviation HRQoL. This corresponds to a scale of the previously mentioned instrument.... It is another instrument?.
If you add this abbreviation, you should explain what you are referring to. For example: Health-Related Quality of Life (HRQoL).
Please, arrange the above paragraph in an appropriate manner.
Response: Thank you very much for your suggestion. We have added in introduction:
Finally, few studies consider concepts from positive psychology, such as health-related quality of life (HRQoL)
And in the instruments section:
The KIDSCREEN-27 is a shortened version of the KIDSCREEN-52 to evaluate HRQoL.
- Data analysis Section.
Use the term "Chi-squared test" instead of "Chi2 test".
Response: done.
- Data analysis Section.
The paragraph in general is written in a confusing way. Write appropriately the type of analysis you used for each of the stated objectives, in the same order in which they are presented in the results.
Response: Thank you for important comment, we have changed by:
The linear relation between the Suicide risk with the self-esteem, HRQoL dimensions and social support was evaluated and described with Pearson’s correlation coefficient. The association between suicide risk with sociodemographic self-harm, psychosocial variables and social-family support including dating violence was evaluated by simple logistic regression, and described with odds ratio (OR) and respective 95% confidence intervals (95% CI). For all the analyses, p < 0.05 was considered statistically significant.
Please, considerer to perform first a univariate logistic regression and, secondly, a multivariate regression to observe a final model to test all predictors simultaneously. Author could do these multivariate regression model for suicidality and for suicide attempt.
Response: The results in Table 4 correspond to a univariate logistic regression analysis. In this study, a multivariate analysis was not performed due to the interest of evaluating the individual association of each variable under study. Although it could be interesting to carry out a mediation analysis in future studies and evaluate the effect of the sum of multiple factors on the suicide attempt variable.
- Categorization criteria for the variable suicidal ideation: the way you categorize the suicidal ideation scale (items 1, 2, 3) should be adequately explained in the methods section. You add it later in the results section (page 6, first paragraph). However, it is important for the reader to understand the criteria used to categorize suicidal ideation before getting to the results. In addition, you should reference what you are relying on to define the cutoff score >5.
In the same vein, it defines “Suicidality risk” as outcome, based on items 1, 2 and 3. However, these items refer specifically to suicidal ideation. Whereas the 4 items of the instrument you use (including item 4: suicide attempt), as a whole, refer to suicidality risk. Please use the precise term to express what items 1, 2 and 3 measure.
Change the term suicidality risk to the appropriate term in the main text.
Response: Done, we committed a mistake based on items 1, 2 and 3. However, these items refer specifically to suicidal ideation. We use the classification criteria of category suicidality risk >5. Scores >5 points define suicidality risk
The concepts: suicide attempts. suicidal ideation suicidality risk
We deleted: Suicidality category was derived from suicidality scale items #1, #2 and #3, reflecting a total score from zero to twelve.
We have adapted the text in material section according the correct concept:
The Suicidality Scale was created by Okasha et al. [15]. It is self-report, and comprised of four items, where the first three explore suicidal ideation, and the fourth questions asks about suicide attempts. The four items are Likert-type, each with four categories. The total score ranges between 0 and 12 points. Scores >5 points define suicidality risk. It has a Cronbach’s alpha coefficient of 0.87, and was validated by Salvo [23].
- Suicide attempts criteria: Nor does it explain in the methods section the criteria used to categorize suicide attempts, which it uses as an outcome in the results and Table 5. Again, the reader learns about this criterion indirectly when reaching the results and Table 5.
On page 7, first paragraph, it adds: "shows variables' association with suicide attempts, captured by item #4 on the suicidality scale". Then, in Table 5, it is deduced that the criterion used was "One attempt vs. 2 or more attempt".
Please explain and justify the criterion used in the method section.
Response: we have added this information:
The Suicidality Scale was created by Okasha et al. It is self-report, and comprised of four items, where the first three explore suicidal ideas “suicide risk”, and the fourth questions asks about su-icide attempts and has three possible answers: a. No attempts, b. One attempt, c. Two attempts, d. Three or more. For statistical analyzes two groups was developed: a. group with one attempt and b. group with the group Two or More Attempts
Why did you use the criterion "One attempt vs. 2 or more attempt"? Isn't it more interesting to compare participants with no suicide attempts vs. 1 or more suicide attempts?
Response: In table 5, the reference category used for the association analysis corresponds to who have never attempted suicide.
we have added at the bottom of the table:
Reference category without suicide attempts.
- All results are shown according to differences by sex, which suggests that exploring these differences in the variables of interest is relevant to the objectives of the study. However, in the objectives stated at the end of the introduction, this objective is not made explicit, nor why it is relevant for the research field of self-harm and suicidality risk to know these differences. It is advisable to include a specific objective at the end of the introduction.
Response: thank you for your comment, we have changed and adapted the objective:
The objective of the present study was to determinate the association between suicide risk with self-harm, psychosocial variables, social-family support, sex and sociodemographic parameters in Chilean adolescents.
- Due to the importance that the authors assign to self-harm as a predictor of suicidal behavior and, as the authors express in the main text, the existing overlap of these two risk behaviors, it is advisable to compare what percentage of the subjects who reported self-harm present suicidal risk and suicidal attempt. This information may be useful to explore and know the overlap of these two behaviors in the sample of adolescents evaluated. This is one of the questions that the literature currently recommends to clarify.
Response: Dear reviewer, in table 4 we have information in relation to the risk in OKASHA: Low (≤ 5 points) *N = 535 (64.54) High (>5 points) N = 294 (35.46), according self-harm yes or not, ad we have estimated the OR considering such as reference value no self-harm and evaluate the association.
Low (≤ 5 points) High (>5 points)
Self-Harm *N = 535 (64.54) N = 294 (35.46),
No 469 (87.66) 138 (46.94) Comparison
Yes 66 (12.34) 156 (53.06) 8.03 (5.69-11.33), <0.001
We have marked this information in the table.
- Table 4: The results are reported in a confusing way, with a lot of information included in the table. For example, it is not clear which is the reference category in the predictors included in the regression analyses. In relation to the outcome (Suicidality risk), it reports the prevalence and % of both levels of the variable. This is informative but the focus of the logistic regression is the odd ratio of the predictor with respect to the outcome. The latter is what the Table should highlight. It is therefore suggested to simplify the table and include the information relevant to the objectives of the manuscript (i.e., odd ratio information).
Response: we have added information to clarified the information about reference category.
* Represent reference category Low (≤ 5 points).
- Table 4 and 5: It only reports univariate logistic regression analyses. It is desirable to also include in the same table multivariate logistic regression analyses. This allows to explore which predictors are more robust in predicting Suicidality risk, when controlling for the other variables in the same analysis.
Guidance on how to perform and report these analyses on a similar topic can be found in the following reference:
Mendez, I., Sintes, A., Pascual, J. C., et al. (2022). Borderline personality traits mediate the relationship between low perceived social support and non-suicidal self-injury in a clinical sample of adolescents. Journal of affective disorders, 302, 204–213. https://doi.org/10.1016/j.jad.2022.01.065
Response: we have reviewed this interesting investigation, but we were not working on mediation models on this occasion, but in future research we will consider it. Initially, this is not a study that evaluates the mediating effect of sociodemographic or psychosocial variables.
- Table 4 and 5: Its should use the same format and style in reporting the data, as it considers the same analysis and predictors.
Response: Dear reviewer we have explained this previously according your interesting report. Thanks again.
Discussion.
- Paragraph 2. “…including studies conducted recently in Chile”.
The cited study was published in 2012 and therefore the prevalence may have varied in the last decade. Avoid the term "recently".
Response: Thanks to your annotation, we checked again the literature and found this reference, which we added it and removed the word "recently", since the reference is a chapter of a book, which although recently was published, only mentions previous studies from our country.
Quijada, Y. Understanding Social Risk Factors in Chilean Adolescent Suicides: An Analysis of Mediating Mechanisms. In: Barce-lata Eguiarte, B.E., Suárez Brito, P. (eds) Child and Adolescent Development in Risky Adverse Contexts. Springer, Cham. 2021, https://doi.org/10.1007/978-3-030-83700-6_8
- Paragraph 4. “This invites us to consider self-harm as a critical risk factor in suicide prevention, in contrast to considering self-harm as something relatively benign, such as a mere cry for attention”.
The literature related to Non-suicidal Self-injury in the last decade has clarified that the popular belief of "Self-harm as a mere cry for attention" is erroneous and that those who self-injure have both personal risk factors (e.g., poor emotional regulation skills and social risk factors (e.g., perceived rejection)) associated with them. If the authors have a citation of scientific research that supports the idea of "such as a mere cry for attention" as a popular idea or as a belief of health professionals, please add, otherwise, omit this idea.
Response: Thank you very much for your suggestion, we will add the citation. We think there was also a translation error.
- Limitations Section.
The limitations are adequately expressed in the paragraph. However, another important limitation is the possible overlap between self-harm and suicidality risk in the analyses conducted by the authors. While previous literature has evidenced this overlap, future studies can explore this dynamic by adequately differentiating non-suicidal self-injury behaviors with suicidal behavior. To the above, the appropriate question to explore NSSI might be "Have you ever self-injured yourself, with no intention of dying?".
It is likely that the question used by the authors ('Have you ever self-harmed?) captured both NSSI and suicidal behaviors in the participants tested. It is advisable to add this limitation.
Response: Thank you very much for your suggestion, we have added:
In this question a history of self-harm was not considered and could be possible overlap between self-harm and risk of suicide. It would also have been advisable to include the question “Have you ever self-harmed, with no intention of dying?", to make it possible to delve into the phenomenon of non-suicidal self-harm.
On the other hand, regarding dating violence, the perception of feeling mistreated by a boyfriend/girlfriend could only be measured with one question, and not with a validated test that measures dating violence, since it was not possible to find a validated questionnaire to study adolescent dating violence in Chile.
Thank you very much again for your suggestions
Best regards

Round 2
Reviewer 3 Report
The article has been improved. However, I think that some further modifications and a significant improvement in formal English aspects need to be made before publication. A check of typing errors should be done along the manuscript (e.g, line 46). Furthermore, a consistent format when presenting numerical results is necessary (e.g, “OR;” or “CI;” are sometimes used instead of “OR=” or “CI=”).
INTRODUCTION
-
When you cite textually (in quotes) the reference number 6 (lines 49-51), what is important is to cite exactly the reference, using the same words ("suicide risk" or "suicidality risk" based on the original reference).
-
I appreciated your effort to adopt the same order when listing the variables associated with suicidality risk in the various parts of the paper. However, I suggest a possible improvement. When you explain the aim of the work (lines 82-84), you may adopt this order and maintain it along the paper (methods, results (including tables), discussion): (1) Sociodemographic variables (including sex and school); (2) Psychosocial variables; (3) Social-family support; (4) Self-Harm. In this way, it could be easier for the reader to follow the development of the work.
METHODS
-
A few references are missing and need to be cited: line 89 (WHO definition of adolescence); line 119 (Rosenberg self-esteem scale); line 126 (Kindscreen 27)
-
Have you considered “Physical and mental health” (line 118) as the “psychosocial variables” mentioned in the introduction (line 83)? If yes, please use the same label “Psychosocial variables” instead of “Physical and mental health”, to be clearer and consistent with the Introduction. If not, please consider changing the Introduction.
-
Do the reported Cronbach’s alfa values refer to your sample or to other clinical or community samples? Please, specify and, if necessary, provide adequate references.
-
“Considers aggression a problem” (line 154) is a grammatical error. Please correct.
RESULTS
-
I would consider to use only one table instead of the first two, given that you present the variables of the sample using the same comparison based on sex
-
Table 2: the variable “Relationship” is presented here for the first time, without the corresponding question presented in the Methods section. Please, add it, to make the paper more consistent in its parts.

Author Response
RESPONSE TO REVIEWER
Dear reviewer, thank you very much for the opportunity to correct the article at this stage, your suggestions have really improved the quality of the article.
The article has been improved. However, I think that some further modifications and a significant improvement in formal English aspects need to be made before publication. A check of typing errors should be done along the manuscript (e.g, line 46). Furthermore, a consistent format when presenting numerical results is necessary (e.g, “OR;” or “CI;” are sometimes used instead of “OR=” or “CI=”).
Response: We have made a new review and corrected what was suggested. Anyway, the article was sent to editing services by Proof-reading service.
INTRODUCTION
When you cite textually (in quotes) the reference number 6 (lines 49-51), what is important is to cite exactly the reference, using the same words ("suicide risk" or "suicidality risk" based on the original reference).
Response: Considering your suggestions, we have added more information to clarify this:
In addition, suicidality is a public health problem that requires attention. Suicidality risk is defined as the risk of suicide, usually indicated by suicidal ideation or intent, especially as evident in the presence of a well-elaborated suicidal plan [6].
I appreciated your effort to adopt the same order when listing the variables associated with suicidality risk in the various parts of the paper. However, I suggest a possible improvement. When you explain the aim of the work (lines 82-84), you may adopt this order and maintain it along the paper (methods, results (including tables), discussion): (1) Sociodemographic variables (including sex and school); (2) Psychosocial variables; (3) Social-family support; (4) Self-Harm. In this way, it could be easier for the reader to follow the development of the work.
Response: Dear reviewer we have adapted the structure in a similar way (only different self-harm):
to determine the association between the risk of suicidality with self-harm, sociodemographic parameters (that is, gender and type of school), psychosocial variables, social and family support in Chilean adolescents.
METHODS
A few references are missing and need to be cited: line 89 (WHO definition of adolescence); line 119 (Rosenberg self-esteem scale); line 126 (Kindscreen 27)
Response: We add the following cites and references:
World Health Organization. Adolescent health. (Reviewed July 2022 year). Available on: https://www.who.int/health-topics/adolescent-health#tab=tab_1).
Rosenberg, Morris. "Rosenberg self-esteem scale (RSE)." Acceptance and commitment therapy. Measures package 61.52 (1965): 18.
Sepúlveda R., Molina T., Molina R., Martínez V., González E., Montaño R., et al. Adaptación transcultural y validación de un instrumento de calidad de vida relacionada con la salud en adolescentes chilenos. Rev Med Chil. 2013, 141 (10), 1283-92.
Have you considered “Physical and mental health” (line 118) as the “psychosocial variables” mentioned in the introduction (line 83)? If yes, please use the same label “Psychosocial variables” instead of “Physical and mental health”, to be clearer and consistent with the Introduction. If not, please consider changing the Introduction.
Response: done, we have changed it, by Psychosocial variables
Do the reported Cronbach’s alfa values refer to your sample or to other clinical or community samples? Please, specify and, if necessary, provide adequate references.
Response: the alpha reported belong to the sample study we have added in text the information.
The Cronbach’s alfa in the present study was
“Considers aggression a problem” (line 154) is a grammatical error. Please correct.
Response: We have changed by; Do you consider aggression a problem?
RESULTS
I would consider to use only one table instead of the first two, given that you present the variables of the sample using the same comparison based on sex
Response: dear reviewer, we have two tables, the tables has been solicited in a previous peer review process. sorry for this.
Table 2: the variable “Relationship” is presented here for the first time, without the corresponding question presented in the Methods section. Please, add it, to make the paper more consistent in its parts.
Response: done, we have changed by Aggression in dating, yes n (%)
Thank you very much again for your suggestions.
Best regards

This manuscript is a resubmission of an earlier submission. The following is a list of the peer review reports and author responses from that submission.
Round 1
Reviewer 1 Report
Review Report
Comments for the author
This manuscript conducted a cross-sectional study: Psychosocial Risk Factors and Sociodemographic Characteristics Associated with Suicidality in Chilean Adolescents, which is an important issue for identifying factors of high risk of suicidality in children adolescents. While there are several unclear descriptions in this manuscript should be concerned and require greater details and re-organized.
1. In abstract, Line 35-36, “suicidality was associated with exposure to physical and psychological aggression between parents”. Please indicate the statistical values.
2. In the introduction, such as Line 67-69, please add more details and highlight the research gaps internationally instead of only focus in Chile, since the audiences of the journal are from all over the world.
3. In Line 76-77, the author described that “studies suggest that low self-esteem is a predictor of suicidal ideation [18], as well as completed suicide [19]. Please discuss more here to present the reasons why this study requires to investigate further. In addition, several factors including familial conflict, relationship difficulties with peers, such as being the victim of dating or other violence, all of which were emphasized as predictors of suicidality. Then please highlight the research gaps which need more study to identify.
4. In method section, please separate the participant paragraph into two sections: “research design” and “participants and settings” and describe the sample size estimation required. In addition, the paragraph of “Procedure” can be integrated into the paragraph of “participants and settings”.
5. I don’t understand the description in the section of “2.3. Suicidality and self-harm”. Did the author mean that the score of Suicidality Scale presents the risks of suicide and the score of Impulsivity Scale presents the behavior of self-harm? If so, please describe clearer.
6. In all of the instrument description, please indicate the cut points of the scores or the meaning when the participants had a higher or lower score.
7. In table 1, the variable: please describe the meaning of “N persons at home”. The author did not mention the variable in the study purpose or in the section of statistical analysis.
8. In the table 4, please use the word “reference” rather than “comparison”.
9. In the results section, the information are comprehensive, however, it seems need to re-organize since the results presentation did not follow the research purpose very well. If the difference of gender on those suicide related variables is most important, the author should describe in both of the introduction section and statistical analysis section. Lots of factors are related to suicidality including variables such as sex, school type, self-esteem, health-related quality of life, social support, impulsivity, parental psychological aggression, parental physical aggression, etc. presented in your table results. However, what are the most important predictors after adjusting for potential covariates. It comes to the major purpose of the study. In my opinions, based on previous univariate analysis, there should be a final table to display the association between the suicide risk and overall significant independent variables (sex, school type, self-esteem, health-related quality of life, social support, impulsivity, parental psychological aggression…) using logistic regression.

Reviewer 2 Report
This was well written
The research design seemed appropriate as did the use of selected statistical measures
Reviewer 3 Report
The manuscript on Psychosocial Risk Factors and Sociodemographic Characteristics Associated with Suicidality in Chilean Adolescents deals with a very sensitive subject, suicidality.
1. Suicidality is defined as deliberately hurting oneself with an intent to die (from: The Handbook of Dialectical Behavior Therapy, 2020). There are no such cases in this study. This demonstrates the need for the authors to define very well, from the very beginning of the manuscript, what they mean by suicidality. Probably they consider includes both suicidal ideation and actual suicide attempts. Since in the reported cases there have been no suicide attempts, but only suicide ideation, it would be better that the title and the text talk about suicide ideation and not suicidality.
2. The authors have a suicidality scale that they attribute to Okasha, but they do not indicate the bibliographic entry relating to the original questionnaire. Authors should therefore provide more information on this tool, quote at least one question, and specify whether this tool serves, as it would appear, to measure suicidal ideation.
3. In addition to information about the original instrument, authors should indicate what the internal consistency (Cronbach's alpha) was in their data.
4. in the Results, Table 1, the authors set a Suicidality Scale and a Suicidal Ideation score. What is this score and where does it come from?
5. In the same Table 1 the authors indicate that they compared the means with Student's t test. This test is not appropriate for non-parametric data, they should have used the Mann-Whitney Wilcoxon U test.
6. In the Results, on lines 191 and following, the authors make an operation whose ratio should have been made explicit in the Methods. They reduce the 4-question scale to just three questions (why?) And arbitrarily set a 5-point cut-off. Then they carry out a series of analyses on this new indicator. This procedure is incorrect. The authors should have correlated the scores of the complete questionnaire of 4 questions and not only 3. If the cut-off is not based on clinical data it is not acceptable.
7. In line 208 the authors give information that should have been in the methods: "Table 5 shows variables’ association with suicide attempts, with suicide attempts captured by item # 4 on the suicidality scale. " Only at this point do they tell us that the questionnaire includes a declaration of attempted suicide, moreover expressed with a four-point Likert scale. This perhaps explains why in the previous evaluation they used only three. But it is also very important to understand something more on this scale, how it is possible to measure the attempted suicide in four points, how these four points are grouped into a YES or a NO, and what value this operation has.
8. In conclusion, the study could be strongly biased by a scale whose original formulation is unknown and whose validity is doubtful.
9. The authors do not indicate how many adolescents who attempted suicide, or rather those who declared that they had attempted suicide, would have been in their sample. The validity of this statement should be debated, also in consideration of the very high percentage of self-harm.